# Hint-AD: Holistically Aligned Interpretability in End-to-End Autonomous Driving

**Kairui Ding**[1,3]**, Boyuan Chen**[1,3]**, Yuchen Su**[3]**, Huan-ang Gao**[1]**, Bu Jin**[1]**, Chonghao Sima**[4]**,**
**Wuqiang Zhang**[2]**, Xiaohui Li**[2]**, Paul Barsch**[2]**, Hongyang Li**[4]**, Hao Zhao**[1,†]

[1]Institute for AI Industry Research (AIR)  [2]Mercedes-Benz Group China Ltd.
[3]Xingjian College, Tsinghua University  [4]OpenDriveLab, Shanghai AI Lab
[†] Indicates corresponding author

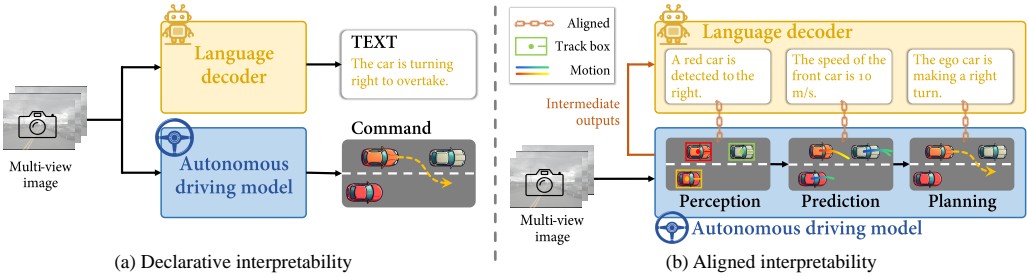

Figure 1: **Illustration of two paradigms for interpretability** of end-to-end autonomous driving (AD) systems through natural language. (a) The *declarative* interpretability does not utilize intermediate outputs from AD systems, resulting in text that merely justifies the car's driving behavior; (b) *Aligned* interpretability incorporates intermediate outputs from the AD model for alignment.

**Abstract:**

End-to-end architectures in autonomous driving (AD) face a significant challenge in interpretability, impeding human-AI trust. Human-friendly natural language has been explored for tasks such as driving explanation and 3D captioning. However, previous works primarily focused on the paradigm of *declarative* interpretability, where the natural language interpretations are not grounded in the intermediate outputs of AD systems, making the interpretations only declarative. In contrast, *aligned* interpretability establishes a connection between language and the intermediate outputs of AD systems. Here we introduce Hint-AD, an integrated AD-language system that generates language aligned with the holistic perception-prediction-planning outputs of the AD model. By incorporating the intermediate outputs and a holistic token mixer sub-network for effective feature adaptation, Hint-AD achieves desirable accuracy, achieving state-of-the-art results in driving language tasks including driving explanation, 3D dense captioning, and command prediction. To facilitate further study on driving explanation task on nuScenes, we also introduce a human-labeled dataset, Nu-X. Codes, dataset, and models are publicly available at `https://air-discover.github.io/Hint-AD/`.

**Keywords:** Interpretability, Autonomous driving, Language alignment

## 1 Introduction

End-to-end perception-planning architecture is critical in autonomous driving (AD) [1, 2] and general embodied intelligence [3, 4] due to its potential for self-supervised training with extensive data. However, these systems face significant interpretability challenges [5, 6].

Interpretability issue [7, 8, 9, 10, 11] is particularly pronounced in embodied intelligence problems such as AD. When an AD system directly outputs control signals, it becomes difficult for human

8th Conference on Robot Learning (CoRL 2024), Munich, Germany.

passengers to trust the decisions. To address this, natural language, a highly user-friendly communication medium, has been explored to enhance interpretability through tasks like driving explanation, 3D dense captioning, and visual question answering (VQA). While human driver recognizes the value of the bird's eye view (BEV) trajectories as an explanation of *WHAT* is happening, language offers a complementary perspective on *WHY* this is happening. These approaches can be categorized into *declarative* interpretability and *aligned* interpretability based on a single criterion: whether the generated language aligns with the intermediate outputs of the AD system (Fig. 1).

- **Declarative interpretability** generates natural language directly without intermediate inputs from the AD system, as in recent works on driving explanation [12, 13, 14], 3D dense captioning [15], and visual question answering [16, 17, 18, 19]. This approach often suffers from hallucination, as the language is not grounded in comprehensive intermediate outputs, making it mere justification of the driving behavior.

- **Aligned interpretability** requires alignment between language and the inner states of the AD model. To our knowledge, this approach was first explored by [14], aligning attention states of the AD model with a language decoder. Later works aligned the language decoder with the inner decision states [20].

However, existing research neglects the correspondence between the language decoder and the complete perception-prediction-planning outputs of an AD pipeline, resulting in a discrepancy between the language tasks and AD tasks. The potential to enhance the accuracy of language tasks in driving scenes through intermediate AD pipeline outputs remains unexplored. To this end, we propose Hint-AD, an integrated AD-language framework designed for holistic alignment with the AD model's perception-prediction-planning process and high-accuracy language generation to facilitate the interpretability in autonomous driving.

We developed two approaches for the holistic alignment between language and the AD model and the accuracy of language outputs: (a) We develop a holistic token mixer module that adapts the intermediate output tokens from the AD model for the language decoder, focusing on robust feature extraction and fusion; (b) We introduce an alignment task as an online dataset to align the language output with the intermediate outputs of the AD model, requiring the language decoder to interpret intermediate tokens generated during the AD model's inference process throughout training.

We implemented Hint-AD on both UniAD [2] and VAD [21], state-of-the-art (SOTA) AD models utilizing rasterized and vectorized representations, respectively, to demonstrate its generality. Experimental results show that Hint-AD achieves state-of-the-art performance on various language tasks, including driving explanation (20.4% on CIDEr than baseline), 3D dense captioning (185% on CIDErthan baseline), VQA (1.2 percent improvement on accuracy), and driving command prediction (1.2 percent improvement of accuracy). The alignment tasks significantly improved coherence between language outputs and intermediate AD model representations. Additionally, we contributed a human-labeled driving explanation dataset, Nu-X on nuScenes [22] to address the lack of driving explanation data on this widely-used AD dataset.

## 2 Related Works

**End-to-end autonomous driving** systems aim to create an architecture that processes sensor data and directly outputs vehicle control signals [23]. These systems have gained research attention due to their ability to address error accumulation problems found in traditional modular designs, where perception and planning are separated into distinct modules [24, 25, 26, 27, 28, 29, 30, 31, 32]. Prominent examples include UniAD [2] and VAD [21] integrate modular perception tasks such as object tracking, map building, motion forecasting, and trajectory planning within a unified framework. Offline datasets for end-to-end autonomous driving have also been developed [22, 33].

**Interpretability of AD** [5, 6, 7, 8], the ability to provide a comprehensive explanation for AD planning, is crucial for user trust and system transparency in AD systems [5, 6]. Natural language, as a

user-friendly medium for communicating with users, has been explored for improving interpretability of AD through by means like driving explanation [12, 13, 14], VQA [16, 17, 18, 19], and 3D dense captioning [15]. Previous work mainly focused on declarative interpretability, For example, [12, 34] realized driving explanation tasks using visual information. But intermediate outputs from AD model is not aligned. [14] raised the concept that the language output should be grounded on the inner states of an AD system. Aligning the decision state between the language decoder and the inner states of AD model has also been explored [35], but to our knowledge, no previous work has achieved holistic alignment with all perception-prediction-planning process with an AD model.

## 3 Methodology

To explore holistic alignment between natural language and intermediate results in end-to-end AD frameworks, we propose a novel framework named Hint-AD, which consists of three modules: a holistic token mixer, a language decoder, and a traditional AD framework. An overview of Hint-AD is shown in Fig. 2. The existing AD pipeline in Fig. 2 can be any end-to-end AD system that decomposes AD into perception, prediction and planning. Without loss of generality, we implement our method on top of both UniAD [2] (as *Hint-UniAD*) and VAD [21] (as *Hint-VAD*), which use rasterized and vectorized representations, respectively.

### 3.1 Overall framework of Hint-AD

**Firstly**, we extract intermediate query tokens from the existing AD model of a perception-prediction-planning architecture, yielding track tokens, motion tokens, and planning tokens. **Secondly**, a holistic token mixer module will adapt the tokens for language decoder input, in which we design an *instance mixer* to merge instance-level track and motion information of each detected instance. We also introduce *BEV blocks* and *instance blocks* for further feature extraction and converting length-variable instance tokens to a fixed length. All the processed tokens are concatenated as context tokens for text generation (see Sec. 3.2). **Finally**, the context tokens are formulated as prompt tokens and put into the language decoder together with text prompts. We adopt a *barbell adaptation* paradigm for efficient context understanding of the language decoder (see Sec. 3.3).

For aligning language and the intermediate results of the AD pipeline in training, we incorporate extra training data called *alignment task*, which is constructed online during training (see Sec. 3.4). Additionally, the training procedures are described in Sec.3.5.

### 3.2 Holistic token mixer

The query tokens extracted from the AD pipeline are not directly understandable to the language decoder. Addressing this, we propose a holistic token mixer architecture. The implementation is slightly different between Hint-UniAD and Hint-VAD in detail. We primarily follow the design of Hint-UniAD, while small adjustments for Hint-VAD are provided in Appendix B.3.

We start by giving denotations for query tokens extracted from the AD pipeline. For a typical perception-prediction-planning AD pipeline, we can extract the following components: *BEV tokens* $F_{\text{bev}} \in \mathbb{R}^{H_b \times W_b \times C}$, where $H_b$, $W_b$, and $C$ are the height, width, and channels of the BEV field. *Track tokens* $\{F_{\text{track}}^i\}_{i=1}^{N_{\text{det}}} \subseteq \mathbb{R}^D$ contain position and past trajectory information of each detected object, where $N_{\text{det}}$ is the number of detected objects, and $D$ is the dimension of the token vector. *Motion tokens* $\{F_{\text{motion}}^i\}_{i=1}^{N_{\text{det}}} \subseteq \mathbb{R}^D$ contain predicted future trajectories of each detected object. *Planning steps* $F_{\text{plan}} \in \mathbb{R}^{T_p \times 2}$ would be the future trajectories predicted by the model.

To effectively merge tokens into an instance level, we design a novel *instance mixer* that integrates the track token $F_{\text{track}}^i$ and the motion token $F_{\text{motion}}^i$ of each detected instance into an instance token $F_{\text{instance}}^i$. This is accomplished through tensor concatenation followed by a Multi-Layer Perceptron (MLP) projector $\mathcal{P}_{\text{instance}}$ to project tokens of $N_{\text{det}}$ detected instances into embeddings with dimension $D_{\text{embed}}$:

$$F_{\text{instance}}^i = \mathcal{P}_{\text{instance}}(\text{Concat}(F_{\text{track}}^i, F_{\text{motion}}^i)), \text{for } i = 1, 2, \ldots, N_{\text{det}}. \tag{1}$$

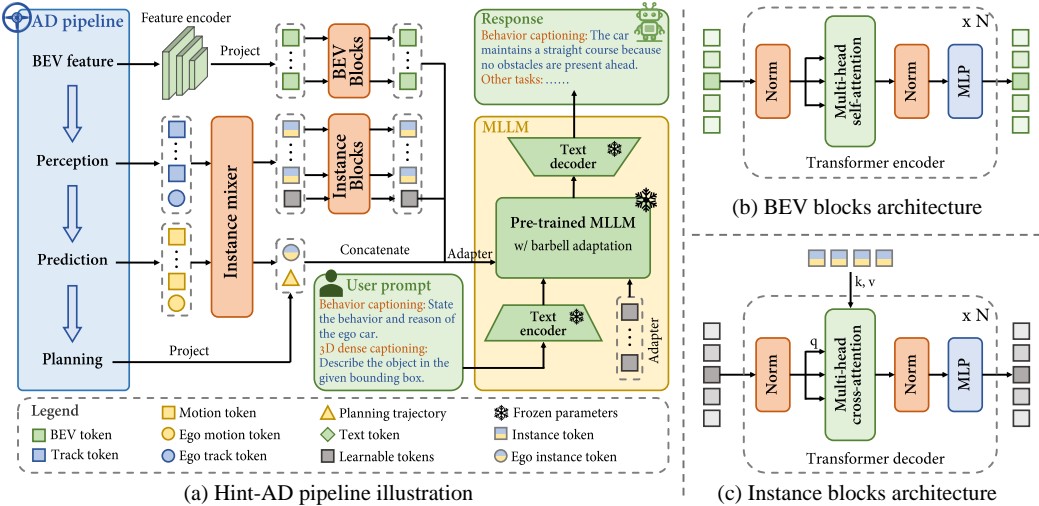

(a) Hint-AD pipeline illustration.     (c) Instance blocks architecture

**Figure 2: Framework of Hint-AD. (a) Hint-AD pipeline illustration. Taking intermediate output tokens from an AD pipeline as input, a language decoder generates natural language responses. A holistic token mixer module is designed to adapt the tokens. (b) Detailed illustration of BEV blocks architecture. (c) A detailed illustration of instance blocks architecture.**

*Feature encoder* $\mathcal{E}$ is implemented as a multi-layer convolutional network, which extracts the feature and down-scales the BEV to $3 \times 3$. Following this, an MLP projector $\mathcal{P}_{\text{bev}}$ is employed to transform the BEV channel dimension $C$ into $D_{\text{embed}}$, yielding $F'_{\text{bev}} = \mathcal{P}_{\text{bev}}(\mathcal{E}(F_{\text{bev}}))$.

*BEV blocks* and *instance blocks* employ multi-head self-attention layers to adapt BEV and instance features. For BEV tokens, multi-head self-attention (MHSA) operates among them. Given the variable number of detected instances per frame, $N_{\text{ins-adapt}}$ learnable tokens $\{F_{\text{ins-adapt}}\}_{i=1}^{N_{\text{ins-adapt}}}$ are introduced as queries. Multi-head cross-attention (MHCA) is then performed between these learnable tokens and the instance tokens. BEV and instance blocks improve adaptation through improving feature extraction and fusion of BEV and instance tokens, as demonstrated in Sec. 4.4.

$$F''_{\text{bev}} = \text{MHSA}(F'_{\text{bev}}), \quad F'_{\text{instance}} = \text{MHCA}(F_{\text{ins-adapt}}, F_{\text{instance}}). \tag{2}$$

The planning steps $F_{\text{plan}}$ would be encoded by sinusoidal positional encoding PE and an MLP projector $\mathcal{P}_{\text{plan}}$ to embedding dimension $D_{\text{embed}}$: $F'_{\text{plan}} = \mathcal{P}_{\text{plan}}(\text{PE}(F_{\text{plan}}))$.

Among all the instances tokens, there's one ego instance token $F_{\text{instance}}^{\text{ego}}$ representing the ego car [2]. Processed BEV, instance, ego instance, and planning tokens would be concatenated as context tokens for further language generation tasks:

$$F_{\text{context}} = \text{Concat}(F''_{\text{bev}}, F'_{\text{instance}}, F_{\text{instance}}^{\text{ego}}, F'_{\text{plan}}). \tag{3}$$

### 3.3 Language decoder with barbell adaptation

To incorporate high-level reasoning and context understanding ability of Multimodal Large Language Models (MLLMs) in AD relevant language tasks [36, 37, 17, 18, 38, 39], we employ a LLaMa-Adapter-V2 [40] as our language generator with a pretrained LLaMA-2-7B [41]. For language fine-tuning, learnable adapters $\{F_{\text{adapter}}^i\}_{i=1}^{N_{\text{adapter}}} \subseteq \mathbb{R}^{D_{\text{dec}}}$ are employed, which serve as extra keys and values in the inserted layers with zero-initialized attention [40], where $N_{\text{adapter}}$ is the number of layers to insert, and $D_{\text{dec}}$ is the dimension of tokens in the language decoder.

In the original LLaMa-Adapter-V2 strategy, context tokens $F_{\text{context}}$ would be inserted in the first layer, and learnable adapters would be inserted in all other $N - 1$ layers, where $N$ is the total number of layers of LLaMA-2-7B. We observed that the adapters, which are for language tuning, tend to dominate the adaptation process and reduce the context-language alignment. This is crucial for AD tasks that requires high-level context understanding ability. Thus we propose a *barbell adaptation*

paradigm (see Fig. 2), where learnable adapters are inserted only at the $[2, N_{\mathrm{front}} + 1]$ layers (as *front* adapters $\{F_{\mathrm{front}}^i\}_{i=1}^{N_{\mathrm{front}}}$) and the $[N - N_{\mathrm{end}} + 1, N]$ layers (as *end* adapters $\{F_{\mathrm{end}}^i\}_{i=1}^{N_{\mathrm{end}}}$). The context tokens are inserted at the first layer.

The rationale for placing adapters at both the front and end is that front adapters aid in comprehending context information, while end adapters enhance language fine-tuning. This design balances the need for high-level context understanding and precise language adaptation. The effectiveness of this barbell adaptation approach is demonstrated in Sec. 4.4. During training, we employ cross-entropy loss as the captioning loss, with supervision applied exclusively to the answer tokens.

### 3.4 Aligning language and intermediate outputs

To align language with intermediate outputs from AD model, the language decoder needs grounded context understanding of the information contained in each token (i.e. object's position in track tokens) generated in AD model's inference steps. We implemented this by adding an online *alignment task* dataset in the training process.

During the alignment task, given the AD model's intermediate inputs, a set of prompt-answer pairs are generated (see Fig. 3). This task includes four types of alignments: (a) **Counting alignment**, requiring the language decoder to interpret the number of instances of each type detected in the frame according to track tokens; (b) **Position alignment**, necessitating the model to provide the position of a tracked instance based on a specific instance token; (c) **Motion alignment**, involving decoding the velocity information contained in an instance token; and (d) **Planning alignment**, requiring the language decoder to output the future trajectory points contained in a planning token.

All the question-answer pairs for alignment tasks are generated online during training. The alignment tasks greatly improve the language decoder's context understanding of the intermediate tokens, thus improving the accuracy of AD captioning by a large margin (see Sec. 4.4).

### 3.5 Training pipeline

The whole training pipeline of Hint-AD includes two stages. In the first stage, the end-to-end AD model is trained independently. In the second stage, we freeze all parameters of the AD model and the MLLM parameters, updating only the parameters of the holistic token mixer and adapters. The total trainable parameters at the second stage are 87M, more training details are stated in Appendix A.

## 4 Experiments

### 4.1 Datasets and baselines

**Datasets.** Explanation serves as a guide for human learning and understanding [42, 43]. Particularly in the context of end-to-end autonomous driving (AD) systems, human users often seek explanations to bridge the gap between sensor inputs and AD behaviors [14]. Currently, there is no dataset providing such explanations for nuScenes [22], a widely utilized dataset in AD research. To address this gap and facilitate interpretability-focused research on nuScenes, we introduce *Nu-X*, a comprehensive, large-scale, human-labeled explanation dataset. *Nu-X* offers detailed contextual information and diverse linguistic expressions for each of the 34,000 key frames in nuScenes.

A sentence of explanation typically comprises narration and reasoning [14], for instance: "<Narration> The car is merging into the right lane. <Reasoning> To pass the red car in front." In our dataset, each caption encompasses these two components. Detailed examples and key data statistics are presented in Fig. 10. For an in-depth description of the labelling process and comprehensive data statistics, please refer to Appendix D.

All Hint-AD architectures and baselines were trained and evaluated on the following datasets to provide a comprehensive analysis: (1) **Alignment task dataset** (see Sec. 3.4), designed to align language with intermediate outputs of the AD model by requiring the language decoder to interpret each intermediate token, with ground truth answers generated online during training; (2) **TOD$^3$Cap** [15], a 3D dense captioning dataset offering object descriptions for 64.3K outdoor objects in nuScenes,

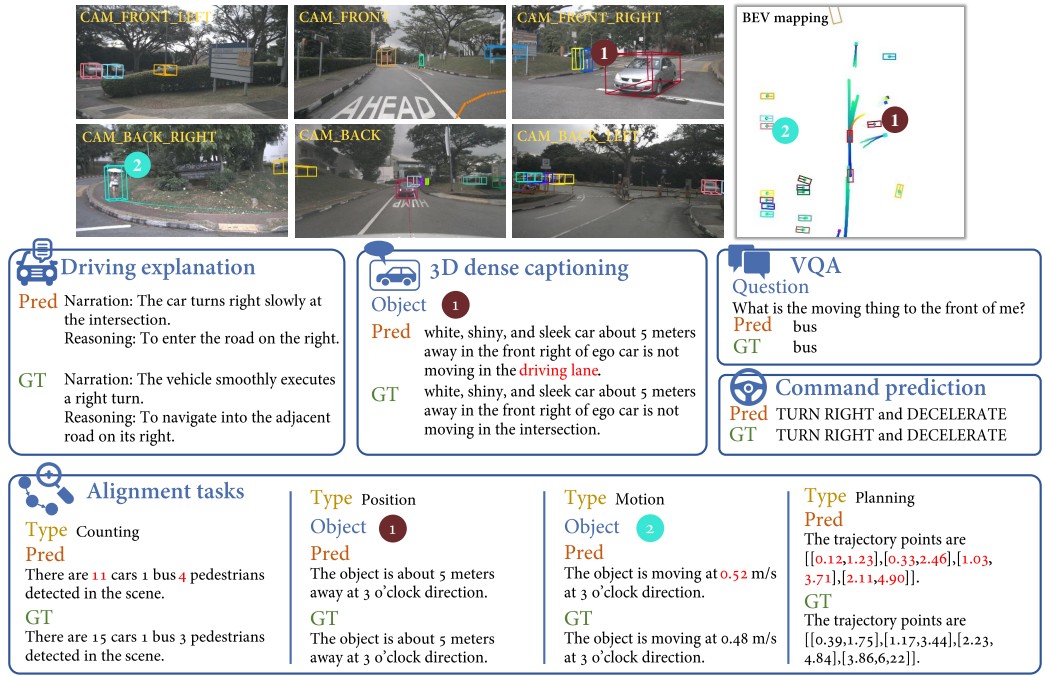

**Figure 3: Qualitative Results.** We present examples of the language output generated by Hint-AD across multiple tasks, including driving explanation, 3D dense captioning, VQA, command prediction, and four categories of alignment tasks. Captions that do not match the ground truth are colored in red.

annotated with appearance, motion, environment, and inter-object spatial relationships; (3) **NuScenes-QA** [16], a VQA dataset covering 34K frames of nuScenes with five question types, including existence, counting, query-object, query-status, and Comparison; (4) **Driving command dataset**, labeled by us on nuScenes (see Appendix E), composed of direction and velocity commands.

**Baselines.** We have selected benchmark methods that include both key milestones and state-of-the-art approaches in language generation within the AD context (more details in Appendix C.4): (1) **ADAPT** [12] generates sentences with a vision-language transformer in an auto-regressive manner. Cross attention and sparse attention masks are used on text and video tokens; (2) **BEV+Adapter** [40] takes only BEV features as input and LLaMA-Adapter-V2 (the same as Hint-AD) as the language decoder; (3) **BEVDet+MCAN** [16] uses a Modular Co-Attention Network (MCAN) [44] with layers of self-attention for separate language and visual understanding. Stacked cross-attention layers are used for cross-model feature interaction. Detection results from a BEVDet [45] is adapted for input; (4) **Vote2Cap-DETR** [46] has two parallel task-specific heads base on a transformer architecture. The queries are decoupled into localization queries and caption queries; (5) **TOD$^3$Cap** [15] utilizes a query-based detection head to generate a set of 3D object proposals from the BEV features. These features are then processed by LLaMA-Adapter [47] to be prompts for the language model to generate dense captions; (6) **GPT-4o** is a multimodal model developed by OpenAI, equipped with state-of-the-art vision capabilities alongside text generation performance comparable to its predecessor, GPT-4. (7) **Gemini-1.5** is a pioneering large language model from Google, specifically designed to handle multimodal inputs with an extended context length.

## 4.2 Comparing with baseline models

**Quantitative results.** We present results separately for different input types and backbone modules on four datasets. For Nu-X and TOD$^3$Cap datasets, we adopt four standard image captioning metrics, CIDEr (C) [48], BLEU (B) [49], METEOR (M) [50] and Rouge (R) [51]. GPT-3.5 scoring (G) is also used for Nu-X' evaluation due to the comprehensive expression in driving explanation (See

Appendix C.6 for detail). A threshold of 0.25 is set for matching predicted and ground-truth bounding box while testing TOD³Cap. For NuScenes-QA and Command datasets, we directly compare the generated texts with ground truth to obtain accuracy. Based on reasoning complexity, QA are divided into zero-hop (H0) and one-hop (H1). Following conclusions can be drawn from Tab. 1.

Both Hint-UniAD and Hint-VAD demonstrate high performance on multi-task tests. Both systems achieve SOTA results on the Nu-X dataset, surpassing the best baseline (BEV+Adapter) by 3.8 points (20.4%) in CIDEr scores. Remarkably, Hint-UniAD exhibits significantly superior performance on the TOD³Cap task, with a CIDEr score improvement of 222.3 points (185%). Although Hint-VAD performs slightly lower on this task, potential explanations are discussed in Appendix C.3. Additionally, on the NuScenes-QA and Command datasets, Hint-VAD achieves overall accuracy improvements of 0.6 and 1.2 points over the best baseline, respectively. These results underscore the effectiveness of the proposed Hint-AD architecture.

Table 1: **Comparison with Baselines.** "Inter. outputs" represents intermediate outputs. All methods are adapted for BEV visual representation and employ mixed dataset training. Hint-UniAD and Hint-VAD, as two implementations of Hint-AD on different AD models, outperform baselines across four language tasks in the AD context.

| Input | Method | Nu-X | | | | | TOD³Cap | | | | NuScenes-QA | | | Command |
|---|---|---|---|---|---|---|---|---|---|---|---|---|---|---|
| | | C ↑ | B ↑ | M ↑ | R ↑ | G ↑ | C ↑ | B ↑ | M ↑ | R ↑ | H0 ↑ | H1 ↑ | All ↑ | Acc. ↑ |
| Image + 6-shot examples | GPT-4o | 19.0 | 3.95 | 10.3 | 24.9 | 5.22 | 160.8 | 50.4 | 31.6 | 43.5 | 42.0 | 34.7 | 37.1 | 75.4 |
| | Gemini 1.5 | 17.6 | 3.43 | 9.3 | 23.4 | 5.03 | 169.7 | 53.6 | 33.4 | 45.9 | 40.5 | 32.9 | 35.4 | 80.9 |
| BEV(2D) | ADAPT [12] | 17.7 | 2.06 | 12.8 | 27.9 | 5.79 | - | - | - | - | 51.0 | 44.2 | 46.4 | 79.3 |
| | BEV+Adapter [40] | 18.6 | 3.47 | 11.3 | 24.5 | 6.27 | - | - | - | - | 51.8 | 45.6 | 47.7 | 81.1 |
| BEV(2D) + bounding boxes | BEVDet+MCAN [44] | 13.2 | 2.91 | 10.3 | 24.5 | 5.04 | 104.9 | 50.1 | 43.0 | 68.0 | **56.2** | 46.7 | 49.9 | 80.7 |
| | Vote2Cap-DETR [46] | 15.3 | 2.61 | 10.9 | 24.2 | 5.33 | 110.1 | 48.0 | 44.4 | 67.8 | 51.2 | 44.9 | 47.0 | 76.5 |
| | TOD³Cap [15] | 14.5 | 2.45 | 10.5 | 23.0 | 5.10 | 120.3 | 51.5 | 45.1 | 70.1 | 53.0 | 45.1 | 49.0 | 78.2 |
| BEV(2D) + inter. outputs | **Hint-UniAD (Ours)** | 21.7 | **4.20** | 12.7 | 27.0 | 7.20 | **342.6** | **71.9** | **48.0** | **85.4** | **56.2** | 47.5 | 50.4 | **83.0** |
| | **Hint-VAD (Ours)** | **22.4** | 4.18 | **13.2** | **27.6** | **7.44** | 263.7 | 67.6 | 47.5 | 79.4 | 55.4 | **48.0** | **50.5** | 82.3 |

**Qualitative Results.** Figure 3 presents some qualitative results. The text generated by Hint-AD demonstrates a grounded understanding of the scene and aligns appropriately with the intermediate results of the AD model. For additional results and analysis, please refer to Appendix C.2.

### 4.3  Analysis on alignment between language and AD model

To quantify the alignment between language and the intermediate outputs of the AD model, we evaluate the language decoders' output against the predictions of the AD perception modules, which is generated online on the validation set. Four kinds of *disalignment* protocols are designed: (a) *Counting Disalignment (CD)* which measures the discrepancy between the number of instances in each category given by the decoding head and the tracking model, (b) *Position Disalignment (PD)* which measures the positional difference of a specific instance, (c) *Motion Disalignment (MD)* which measures the velocity difference, calculated as the mean distance between the velocity in the caption and the velocity predicted by the perception system, and (d) *Planning Disalignment (PLD)* which measures the discrepancy in trajectory points. For detailed definition, please refer to Appendix C.1.

We tested Hint-AD with both aligned interpretability (original design) and declarative interpretability, results are detailed in Appendix C.1. The aligned language decoder performs significantly better than the models operating under the declarative interpretability paradigm, indicating the effectiveness of alignment designs including holistic token mixer and alignment tasks.

### 4.4  Ablation study

**Effectiveness of holistic alignment.** To evaluate the effectiveness of holistic language-AD alignment on language task accuracy, we conducted an ablation study by removing track, motion, and planning tokens from the language decoder's inputs. Results in Tab. 2 show that using all tokens achieves the highest performance. Track tokens enhance 3D dense captioning with positional information, while planning tokens improve command prediction by providing future trajectory data.

Table 2: **Ablation on holistic alignment.** The performance is highest when all tokens are used, highlighting their importance in enhancing specific tasks.

| Method | Input | Nu-X | | | | | TOD³Cap | | | | NuScenes-QA | | | Command |
|---|---|---|---|---|---|---|---|---|---|---|---|---|---|---|
| | | C ↑ | B ↑ | M ↑ | R ↑ | G ↑ | C ↑ | B ↑ | M ↑ | R ↑ | H0 ↑ | H1 ↑ | All ↑ | Acc. ↑ |
| Hint-UniAD | W/o track token | 19.5 | 2.95 | 11.3 | 21.3 | 6.85 | 120.3 | 48.0 | **48.5** | 69.4 | 53.0 | 42.6 | 46.1 | 82.5 |
| | W/o motion token | **22.3** | 3.02 | 12.6 | 26.5 | 7.04 | 267.3 | 67.2 | 43.0 | 81.3 | 53.6 | 43.5 | 46.9 | 82.4 |
| | W/o planning token | 20.8 | 3.10 | 11.4 | 23.5 | 6.39 | 290.6 | 68.4 | 44.5 | **85.6** | 55.3 | 46.7 | 49.6 | 79.1 |
| | w/ all tokens | 21.7 | **4.20** | **12.7** | **27.0** | **7.20** | **342.6** | **71.9** | 48.0 | 85.4 | **56.2** | **47.5** | **50.4** | **83.0** |

**Ablation on Holistic Token Mixer Design.** Instance mixer and instance blocks enhance the feature extraction and adaptation of intermediate tokens. Results in Tab. 3 indicate that removing instance blocks and instance mixer significantly reduces performance on TOD³Cap and NuScenes-QA, as the positional and motion information of objects is not adequately fused.

Table 3: **Ablation on holistic token mixer.** The performance is highest when all sub-networks are included, showing the importance of these components for effective feature extraction and adaptation.

| Method | Sub-networks | Nu-X | | | | | TOD³Cap | | | | NuScenes-QA | | | Command |
|---|---|---|---|---|---|---|---|---|---|---|---|---|---|---|
| | | C ↑ | B ↑ | M ↑ | R ↑ | G ↑ | C ↑ | B ↑ | M ↑ | R ↑ | H0 ↑ | H1 ↑ | All ↑ | Acc. ↑ |
| Hint-UniAD | W/o instance mixer | 20.8 | 3.22 | 10.6 | 26.5 | 6.38 | 220.4 | 58.3 | 46.9 | 73.0 | 52.0 | 44.8 | 47.2 | 81.6 |
| | W/o instance blocks | 21.3 | 3.24 | 11.3 | **27.1** | 7.13 | 259.4 | 62.5 | 47.0 | 82.3 | 54.3 | 47.2 | 49.6 | 81.3 |
| | All sub-networks | **21.7** | **4.20** | **12.7** | 27.0 | **7.20** | **342.6** | **71.9** | **48.0** | **85.4** | **56.2** | **47.5** | **50.4** | **83.0** |

**Effectiveness of barbell adaptation.** We explore three alternatives: (a) *Early Fusion*, original design of LLaMA-Adapter-V2 [40], adapters in each layer; (b) *Pyramid adaptation*, adapters in the first $N_{\text{front}}$ layers; and (c) *Hammer adaptation*, adapters in final $N_{\text{end}}$ layers only. Barbell adaptation performs the best on three datasets (Tab. 4). More adaptation methods in Appendix B.2.

Table 4: **Ablation on adaptation strategy.** Ballbell adaptation achieves the best scores on all datasets except for TOD³Cap, demonstrating our adaptation strategy enables better context understanding and language fine-tuning.

| Method | Adaptation Strategy | Nu-X | | | | | TOD³Cap | | | | NuScenes-QA | | | Command |
|---|---|---|---|---|---|---|---|---|---|---|---|---|---|---|
| | | C ↑ | B ↑ | M ↑ | R ↑ | G ↑ | C ↑ | B ↑ | M ↑ | R ↑ | H0 ↑ | H1 ↑ | All ↑ | Acc. ↑ |
| Hint-UniAD | Early fusion | 16.9 | 3.34 | 12.0 | 25.6 | 6.94 | **350.4** | 70.6 | 45.3 | **89.3** | 55.9 | 47.0 | 49.9 | 78.3 |
| | Pyramid adaptation | 15.3 | 3.10 | 11.2 | 24.4 | 6.34 | 335.3 | 70.8 | 44.4 | 67.8 | 52.5 | 44.6 | 48.5 | 77.5 |
| | Hammer adaptation | 14.7 | 2.45 | 10.5 | 23.0 | 6.19 | 298.7 | 65.6 | 42.1 | 60.1 | 50.4 | 43.1 | 45.5 | 76.2 |
| | Barbell adaptation | **21.7** | **4.20** | **12.7** | **27.0** | **7.20** | 342.6 | **71.9** | **48.0** | 85.4 | **56.2** | **47.5** | **50.4** | **83.0** |

## 5   Conclusions and Limitations

We present Hint-AD, an integrated AD-language framework that aligns language generation with the holistic perception-prediction-planning process of AD models, achieving SOTA performance in multiple AD captioning tasks. Meanwhile, as an exploratory research on the implementation of aligned interpretability, the following limitations are waiting to be resolved by further research:

- Due to its pipeline-specific nature, any changes in the intermediate output format necessitate modifications in the design of the token mixer. For purely end-to-end models, such as black-box models, adjustments are required to handle latent outputs effectively.

- The LLaMA-based language decoder is relatively time-consuming. Further investigation into smaller model alternatives, such as MiniChat-1.5-3B and StableLM-3B-4E1T, is warranted.

As LLM's potential to comprehend AD models' intermediate outputs becomes evident, future research can delve deeper into this area and enhance user trust in AD models through aligned interpretability.

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

# A Training and inference

The graphics memory usage for the language decoder varies with batch size, ranging from 31GB (batch size = 1) to 78GB (batch size = 24). For Hint-VAD, the memory usage ranges from 26GB (batch size = 1) to 45GB (batch size = 20). Training Hint-UniAD takes 13 hours for 3 epochs on 8 A100 GPUs, while training Hint-VAD takes 7 hours on 8 A100 GPUs. The inference times for the AD module and language decoder are detailed in Tab. 5.

Table 5: **Inference Time and Average Time per Batch.** Evaluated on 1 single A100 GPU, this table presents the inference times and average time per batch for the AD model and language decoder of Hint-UniAD and Hint-VAD under different batch sizes of sentences to generate.

| Method | Module | Batch=1 | | Batch=10 | | Batch=20 | |
|---|---|---|---|---|---|---|---|
| | | time (s) | Avg. time/batch (s) | time (s) | Avg. time/batch (s) | time (s) | Avg. time/batch (s) |
| Hint-UniAD | AD model | 0.43 | 0.43 | 0.43 | 0.43 | 0.43 | 0.43 |
| | Language decoder | 1.23 | 1.23 | 1.38 | 0.138 | 1.45 | 0.073 |
| Hint-VAD | AD model | 0.159 | 0.159 | 0.159 | 0.159 | 0.159 | 0.159 |
| | Language decoder | 1.25 | 1.25 | 1.42 | 0.142 | 1.48 | 0.074 |

In Tab. 6, we present the wall-clock time of the entire system with a batch size of 1, as applied in a real car scenario. This time encompasses data acquisition, pre-processing, and post-processing, all measured in seconds.

Data acquisition time includes capturing images from six HIKROBOT MV-CU013-A0UC cameras and transmitting these images from the computer to the online server if using online inference.

Pre-processing time involves preparing image sequences before inputting them into the model, which is conducted online if using an online server.

Autonomous driving inference time refers to the time taken by UniAD and VAD to perform inferences.

LLM inference time is calculated as the time required to generate 26.5 tokens, which is the average generation length observed in our research.

Post-processing time includes the duration needed for data transmission if the inference is conducted online.

Since users can access language output as soon as the first token is generated, we define interactive latency as the sum of data acquisition, pre-processing, autonomous driving inference, and post-processing times.

Table 6: **Whole system latency measurements. Acq. refers to acquisition, AD refers to the inference time of AD model, Token gen. refers to the token generation speed, Inter. refers to interactive.**

| Methods | GPU | Data acq. (s) | Pre-process (s) | AD (s) | LLM (s) | Token gen. (/s) | Post-process (s) | Sum (s) | Inter. latency (s) |
|---|---|---|---|---|---|---|---|---|---|
| Hint-UniAD | RTX 3090 (offline) | <0.01 | <0.01 | 0.49 | 1.83 | 14.5 | <0.01 | 2.32 | 0.49 |
| Hint-UniAD | RTX 4090 (offline) | <0.01 | <0.01 | 0.27 | 0.62 | 42.9 | <0.01 | 0.89 | 0.27 |
| Hint-UniAD | NVIDIA A100 (online) | 0.35 | <0.01 | 0.56 | 1.24 | 21.5 | 0.04 | 2.18 | 0.94 |
| Hint-VAD | RTX 3090 (offline) | <0.01 | <0.01 | 0.12 | 1.65 | 16.1 | <0.01 | 1.77 | 0.12 |
| Hint-VAD | RTX 4090 (offline) | <0.01 | <0.01 | 0.07 | 0.59 | 45.2 | <0.01 | 0.66 | 0.07 |
| Hint-VAD | NVIDIA A100 (online) | 0.35 | <0.01 | 0.16 | 1.25 | 21.3 | 0.04 | 1.79 | 0.54 |

# B More on Model Designs

## B.1 Parameters in Model Design

Here we provide Tab. 7 to specify parameters used in our architecture and additional explanation.

BEV blocks consist of a sequence of 8 transformer encoder blocks. Each block includes a normalization layer, a multi-head self-attention layer with 16 attention heads, a second normalization layer, and a multi-layer perceptron (MLP) with a hidden dimension of 3072.

Instance blocks comprise a sequence of 5 transformer decoder blocks with the same internal structure as the BEV blocks. These blocks accept 5 learnable tokens, $F_{\text{ins-adapt}}$, and instance tokens, $F_{\text{instance}}$, as input. A cross-attention mechanism is performed between these inputs, with $F_{\text{ins-adapt}}$ serving as the queries and $F_{\text{instance}}$ serving as the keys and values.

Table 7: **Notations.**

| Notation | Shape & Params | description |
|---|---|---|
| $N_{\text{front}}$ | 12 | layers of front adapters inserted in LLaMA |
| $N_{\text{end}}$ | 8 | layers of end adapters inserted in LLaMA |
| $D_{\text{embed}}$ | 728 | embed dimension used in holistic token mixer |
| $C$ | 256 | dimension of the BEV feature |
| $F_{\text{bev}}$ | $200 \times 200 \times 256$ | BEV tokens |
| $F_{\text{track}}$ | $N_{\text{det}} \times 256$ | Track tokens |
| $F_{\text{motion}}$ | $N_{\text{det}} \times 256$ | Motion tokens |
| $F_{\text{plan}}$ | $N_{\text{det}} \times 256$ | Planning steps |
| $F_{\text{instance}}$ | $N_{\text{det}} \times 256$ | Instance token |
| $F_{\text{instance}}^{\text{ego}}$ | 256 | Ego instance token |

## B.2  Comparison with Other Adaptation Methods

Here we present analysis on why adapter is chosen as tuning method for Hint-AD.

We selected adapters due to considerations of training stability and parameter efficiency. As highlighted by LLaMA-Adapter [59], parameter-efficient fine-tuning methods like LoRA exhibit instability during training, particularly when addressing entirely new modalities. For instance, LoRA experiences gradient explosion within approximately 2.3K training steps. The LLaMA-Adapter addresses this issue by introducing zero-initialized cross-attention weights between adapter tokens and text queries.

We show experiment results with LoRA and DoRA in Tab. 8, demonstrating that the adapter is the only method that ensures stable training for this application while delivering superior overall performance.

Table 8: **Comparison between adapter and LoRA on Hint-AD.**

| Method | # Params (%) ↓ | Stable training steps ↑ | Nu-X | | | | TOD³Cap | | | | NuScenes-QA | | | Cmd. |
|---|---|---|---|---|---|---|---|---|---|---|---|---|---|---|
| | | | C ↑ | B ↑ | M ↑ | R ↑ | C ↑ | B ↑ | M ↑ | R ↑ | H0 ↑ | H1 ↑ | All ↑ | Acc. ↑ |
| LoRA (r=16) | 0.672 | 2.3K | 17.8 | 3.05 | 10.1 | 21.8 | 257.6 | 61.3 | 24.4 | 57.5 | 45.3 | 40.0 | 41.2 | 71.6 |
| DoRA (r=16) | 0.702 | 3.1K | 18.3 | 3.72 | 10.4 | 21.5 | 290.4 | **69.0** | 37.1 | 64.3 | **52.7** | 42.7 | 46.0 | **78.9** |
| Adapter_epoch-3.1K | 0.014 | **>20K** | **20.8** | **3.90** | **12.9** | **26.5** | **307.3** | 68.9 | **43.3** | **75.6** | 52.4 | **45.5** | **47.8** | 75.6 |

## B.3  Variations on Hint-VAD framework

VAD [21] uses the same BEV encoder as UniAD [2], but vectorizes scene representations for planning and getting rid of dense maps. So the *BEV tokens* are the same: $F_{\text{bev}} \in \mathbb{R}^{H_b \times W_b \times C}$, where $H_b$, $W_b$, and $C$ are the height, width, and channels of the BEV field. But for VAD has different motion prediction and planning phases, so the input tokens are adjusted as follows: In *Track tokens* $\{F_{\text{track}}^i\}_{i=1}^N \subseteq \mathbb{R}^D$ and *Motion tokens* $\{F_{\text{motion}}^i\}_{i=1}^N \subseteq \mathbb{R}^D$, $N$ is the number of agent queries, not the number of detected objects. We select valid tokens based on the query's classification score, using a threshold of 0.5, thereby only considering queries with a high probability of corresponding to actual objects. Additionally, VAD considers both ego-agent interaction and ego-map interaction. We use updated queries as *Ego token* $F_{\text{ego}}^i \subseteq \mathbb{R}^D$, which contain both dynamic and static information of the driving scene. Finally, we adopt *Planning steps* $F_{\text{plan}} \in \mathbb{R}^{T_p \times 2}$, the future trajectories predicted by the model. We adjust the token mixer to accommodate the shape of each token while maintaining the same overall architecture.

## C  More on experiments

### C.1  More analysis on language-AD model alignment

To quantify the alignment between language and the intermediate outputs of the AD model, we evaluate the output of language decoders against the results predicted by the AD perception modules, which is generated online on the validation set. We define four kinds of disalignment as follows:

*Counting Disalignment (CD)* measures the discrepancy between the number of instances of each category given by the text and the number detected by the AD model. It is defined as: $\mathrm{CD} = \sqrt{\frac{1}{N_{\mathrm{cat}}} \sum_{i=1}^{N_{\mathrm{cat}}} (\mathbf{c}_{\mathrm{text}}^{(i)} - \mathbf{c}_{\mathrm{AD}}^{(i)})^2}$, where $N_{\mathrm{cat}}$ is the number of object categories, $\mathbf{c}_{\mathrm{text}}^{(i)}$ and $\mathbf{c}_{\mathrm{AD}}^{(i)}$ are the counts of the $i$th category given by the text and the AD model, respectively.

*Position Disalignment (PD)* measures the difference in the position of a specific instance given by the text and the AD model. It is quantified by calculating the mean distance between the position in the caption $\mathbf{r}_{\mathrm{text}}$ and the position predicted by the AD model $\mathbf{r}_{\mathrm{AD}}$: $\mathrm{PD} = \mathrm{Mean}(|\mathbf{r}_{\mathrm{text}} - \mathbf{r}_{\mathrm{AD}}|)$.

*Motion Disalignment (MD)* measures the difference in velocity, calculated as the mean distance between the velocity in the caption $\mathbf{v}_{\mathrm{text}}$ and the velocity predicted by the AD model $\mathbf{v}_{\mathrm{AD}}$: $\mathrm{MD} = \mathrm{Mean}(|\mathbf{v}_{\mathrm{text}} - \mathbf{v}_{\mathrm{AD}}|)$.

*Planning Disalignment (PLD)* measures the discrepancy in trajectory points. A trajectory is expressed as four future positions of the ego car with 0.5s time steps between each $[\mathbf{r}^{(1)}, \mathbf{r}^{(2)}, \mathbf{r}^{(3)} \mathbf{r}^{(4)}]$, and PLD is defined as the mean distance from the original AD prediction at 2s (L2 loss): $\mathrm{PLD} = \mathrm{Mean}(|\mathbf{r}_{\mathrm{text}}^{(4)} - \mathbf{r}_{\mathrm{AD}}^{(4)}|)$.

Here we offer a comparison of alignment between Hint-UniAD and various declarative alignment methods, as shown in Tab. 9.

- Among the declarative methods, the Hint-UniAD architecture achieved the highest performance, except in the counting disalignment task, demonstrating its overall efficacy.

- Methods utilizing a pre-trained LLM as a language decoder performed poorly on the counting task. This poor performance arises from a bias towards certain numbers: for example, LLaMA tends to output specific integers like 10 and 20.

Table 9: **Comparison with other declarative alignment models.** To standardize the output formats of the language decoders used in these methods, we also tuned the models accordingly. N refers to no, and Y refers to yes.

| Method | Alignment type | LLM | Counting ↓ | Position ↓ | Motion ↓ | Planning ↓ |
|---|---|---|---|---|---|---|
| Hint-UniAD | aligned | Y | **2.36** | **1.22** | **0.51** | **2.98** |
| Hint-UniAD | declarative | Y | 5.82 | 10.01 | 1.98 | 10.12 |
| ADAPT | declarative | N | 8.30 | 14.23 | 2.49 | 13.07 |
| BEVDet+MCAN | declarative | N | 4.69 | 13.05 | 2.18 | 12.53 |
| Vote2Cap-DETR | declarative | N | 5.03 | 11.60 | 2.50 | 12.85 |
| TOD³Cap | declarative | Y | 7.10 | 12.13 | 2.17 | 10.60 |

### C.2  More Qualitative results

Here we demonstrate more detailed Qualitative results on all four datasets. All predictions are generated by Hint-UniAD model, and we choose TOD³Cap as baseline while evaluating results on Nu-X and Command datasets.

Our model excels in generating accurate and contextually appropriate predictions across multiple tasks. For driving explanation and command prediction, our model outperforms the baseline by providing more accurate narration, reasoning, and driving commands. In VQA tasks, it excels in accurately describing the surroundings, including the type and number of objects. In 3D dense captioning, our

model successfully detects surrounding objects and provides precise status descriptions. However, it occasionally make errors in identifying object colors and estimating distances. In alignment tasks, our model performs well across all directions but struggles with precise positional information due to the limitations of image-based data. Additionally, it tends to undercount objects in counting tasks.

We also present some qualitative results from prompting GPT-4o and Gemini-1.5, as shown in Figures 4 to 6. Hint-AD demonstrates better consistency with ground-truth annotations, achieving higher accuracy in VQA tasks such as determining whether an object is moving. In contrast, GPT-4o and Gemini-1.5 tend to provide more diverse descriptions of the scene and driving behavior. The full prompts are shown in Figure 8.

### C.3   Analysis on testing difference between Hint-UniAD and Hint-VAD

According to Tab. 1, VAD performs much worse than UniAD on TOD$^3$Cap dataset but slightly better on the other three datasets. This can be explained by VAD's restricted perception range of 60m by 30m, whereas objects in TOD$^3$Cap often fall outside this range. Since all questions in TOD$^3$Cap are object-centered, missed detections significantly lower the scores on this dataset.

### C.4   Baseline implementations

**ADAPT. [12]**   The original work [12] takes a sequence of raw video frames as inputs, and outputs natural language narration and reasoning. It employs Video Swin Transformer [52] as the video encoder. Two tasks, Driving Caption Generation (DCG) and Control Signal Prediction (CSP), are jointly trained based on the same video tokens.

Here we adopt its text generation head, where the sentences are generated by vision-language transformer in an auto-regressive manner. The text inputs are tokenized and padded to a fixed length, and then are embedded with a segment embedding method. While training, we use cross attention mask within text tokens and sparse attention mask between text tokens and BEV features. The vision-language transformer encoder starts with a "[CLS]" token and generates one word token at a time, consuming previously generated tokens as the inputs until it outputs the ending token "[SEP]" or reaches the maximum length.

**TOD$^3$Cap. [15]**   The original TOD$^3$Cap framework is implemented using a BEV detector module that processes both 2D and 2D+3D inputs. We adapted TOD$^3$Cap for UniAD and VAD by adding an extra captioning head. The inputs for Image BEV features and object proposals are derived from the BEV tokens and tracking tokens of these two AD models, respectively.

**Vote2Cap-DETR++. [46]**   Unlike the traditional "detect-then-describe" methods that first detect objects and then describe them, Vote2Cap-DETR++ uses a transformer-based framework to parallelly decode object localization and caption generation tasks. It uses 3D positional encoding is added to enhance the context-awareness of the generated captions: Absolute Position Token is injected into the caption prefix to indicate the spatial location of the object in the 3D scene and Rank-Based Position Encoding is applied to local context tokens.

### C.5   More Ablation Experiments

We have expanded the ablation studies to include multi-module ablation and cross-model consistency, encompassing both Hint-UniAD and Hint-VAD. The ablation results for Hint-UniAD are presented in Tab. 10 and Tab. 11. These results indicate a consistent performance trend between Hint-UniAD and Hint-VAD. Notably, during multi-module ablation, the performance of the model experiences a further decline.

CAM_FRONT SERIES

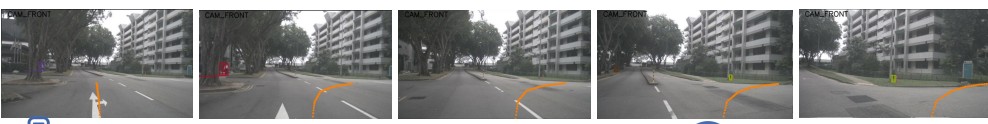

### 🚌 Driving explanation

**Baseline**
Narration: The car continues straight ahead at a steady pace.
Reasoning: because the road ahead is clear of any obstacles.
**Prediction**
Narration: The car turns right at the intersection.
Reasoning: to enter the road on the right.
**GPT-4o**
Narration: The ego car gradually transitions to the right lane.
Reasoning: to prepare for a right turn at the upcoming intersection..
**Gemini 1.5 Pro**
Narration: The ego car is making a right turn onto a side street.
Reasoning: the absence of any other vehicles suggests the ego car is not merely changing lanes but is instead navigating a turn.
**GT**
Narration: The car cautiously navigates a right turn.
Reasoning: to enter the road on the right, maintaining safety guidelines.

### 🎯 Command prediction

**Baseline**
Forward and keep speed.

**Prediction**
Turn right and keep speed.

**GPT-4o**
Turn right and keep speed.

**Gemini 1.5 Pro**
Turn right and keep speed.

**GT**
Turn right and keep speed.

CAM_FRONT SERIES

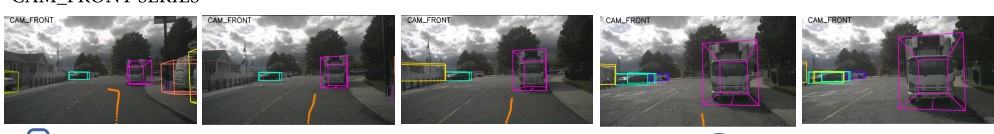

### 🚌 Driving explanation

**Baseline**
Narration: The car stops at the intersection.
Reasoning: because the traffic light is red.
**Prediction**
Narration: The car stops at the intersection.
Reasoning: to keep a safe distance.
**GPT-4o**
Narration: The ego car gradually decelerates and comes to a complete stop.
Reasoning: due to the presence of a large vehicle (likely a bus or truck) directly ahead, which is obstructing the path and necessitating a halt to avoid a collision.
**Gemini 1.5 Pro**
Narration: The ego car is approaching a parked truck and slows down.
Reasoning: The truck remains stationary across the frames, indicating it's parked. As the ego car gets closer to the truck, it's plausible that the car slightly reduces speed to navigate around it safely.
**GT**
Narration: The ego vehicle gradually reduce its speed.
Reasoning: to safely navigate around the parked truck.

### 🎯 Command prediction

**Baseline**
Forward and stationary.

**Prediction**
Forward and stationary.

**GPT-4o**
Forward and decrease speed.

**Gemini 1.5 Pro**
Forward and keep speed.

**GT**
Forward and stationary.

Figure 4: **Qualitative results on Nu-X and Command datasets.** We choose TOD³Cap as baseline model and present GPT-4o and Gemini-1.5 result.

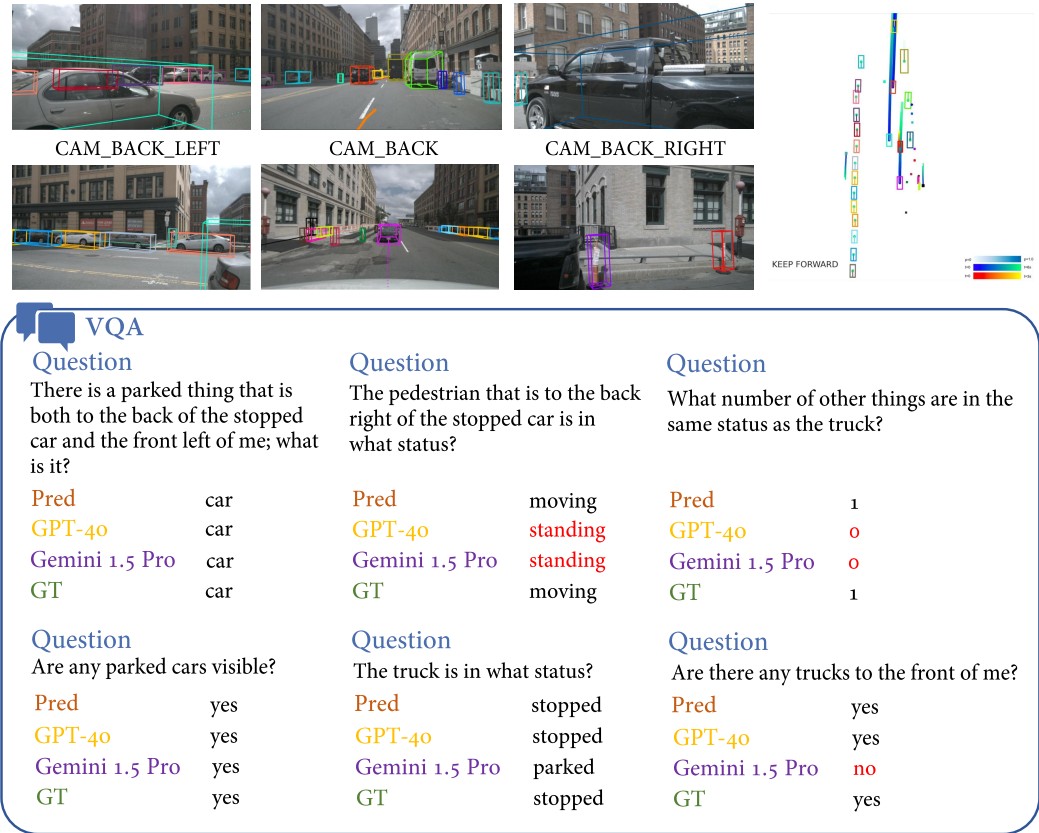

Figure 5: **Qualitative results on NuSence-QA with GPT-4o and Gemini-1.5 outputs.**

Table 10: **More ablation on holistic alignment**

| Method | Ablated | Nu-X | | | | TOD³Cap | | | | NuScenes-QA | | | Cmd. |
|---|---|---|---|---|---|---|---|---|---|---|---|---|---|
| | | C ↑ | B ↑ | M ↑ | R ↑ | C ↑ | B ↑ | M ↑ | R ↑ | H0 ↑ | H1 ↑ | All ↑ | Acc. ↑ |
| Hint-UniAD | track | 19.5 | 2.95 | 11.3 | 21.3 | 120.3 | 48.0 | **48.5** | 69.4 | 53.0 | 42.6 | 46.1 | 82.5 |
| | motion | **22.3** | 3.02 | 12.6 | 26.5 | 267.3 | 67.2 | 43.0 | 81.3 | 53.6 | 43.5 | 46.9 | 82.4 |
| | planning | 20.8 | 3.10 | 11.4 | 23.5 | 290.6 | 68.4 | 44.5 | **85.6** | 55.3 | 46.7 | 49.6 | 79.1 |
| | track & planning | 19.8 | 2.98 | 11.7 | 22.9 | 135.4 | 60.4 | 46.1 | 72.2 | 53.4 | 43.1 | 46.5 | 80.6 |
| | motion & planning | 21.0 | 3.06 | 12.1 | 25.2 | 280.7 | 66.1 | 44.3 | 78.1 | 54.6 | 45.1 | 48.2 | 80.7 |
| | track & motion | 20.1 | 3.03 | 11.9 | 24.1 | - | - | - | - | 54.1 | 44.2 | 47.5 | 81.2 |
| | all | 18.6 | 3.47 | 11.3 | 24.5 | - | - | - | - | 51.8 | 45.6 | 47.7 | 81.1 |
| | none | 21.7 | **4.20** | **12.7** | **27.0** | **342.6** | **71.9** | 48.0 | 85.4 | **56.2** | **47.5** | **50.4** | **83.0** |
| Hint-VAD | track | 20.0 | 3.03 | 11.4 | 22.0 | 132.6 | 44.5 | 46.9 | 56.8 | 52.6 | 46.5 | 48.9 | **83.9** |
| | motion | 19.5 | 2.96 | 11.0 | 21.1 | 233.5 | 59.8 | 44.2 | 71.6 | 54.1 | **49.2** | **50.9** | 81.2 |
| | planning | 18.8 | 2.86 | 10.5 | 20.1 | 246.7 | 66.8 | **47.7** | **78.8** | 53.9 | 47.4 | 49.6 | 77.8 |
| | none | **22.4** | **4.18** | **13.2** | **27.6** | **263.7** | **67.6** | 47.5 | 79.4 | **55.4** | 48.0 | 50.5 | 82.3 |

Table 11: **More ablation on holistic token mixer.**

| Method | Ablated | Nu-X | | | | TOD³Cap | | | | NuScenes-QA | | | Cmd. |
|---|---|---|---|---|---|---|---|---|---|---|---|---|---|
| | | C ↑ | B ↑ | M ↑ | R ↑ | C ↑ | B ↑ | M ↑ | R ↑ | H0 ↑ | H1 ↑ | All ↑ | Acc. ↑ |
| Hint-UniAD | instance mixer | 20.8 | 3.22 | 10.6 | 26.5 | 220.4 | 58.3 | 46.9 | 73.0 | 52.0 | 44.8 | 47.2 | 81.6 |
| | instance blocks | 21.3 | 3.24 | 11.3 | **27.1** | 259.4 | 62.5 | 47.0 | 82.3 | 54.3 | 47.2 | 49.6 | 81.3 |
| | all | 19.1 | 3.02 | 11.1 | 25.1 | 164.3 | 55.1 | 45.1 | 70.2 | 51.1 | 44.1 | 46.4 | 82.1 |
| | none | **21.7** | **4.20** | **12.7** | 27.0 | **342.6** | **71.9** | **48.0** | **85.4** | **56.2** | **47.5** | **50.4** | **83.0** |
| Hint-VAD | instance mixer | 21.6 | 4.09 | 12.4 | 25.8 | 189.4 | 49.3 | 43.8 | 68.3 | 53.0 | 45.1 | 47.7 | 80.7 |
| | instance blocks | 22.0 | 4.17 | **13.3** | 27.3 | 249.8 | 57.9 | 45.7 | 74.5 | 53.6 | 46.1 | 48.6 | 81.0 |
| | none | **22.4** | **4.18** | 13.2 | **27.6** | **263.7** | **67.6** | **47.5** | **79.4** | **55.4** | **48.0** | **50.5** | **82.3** |

CAM_FRONT_LEFT CAM_FRONT CAM_FRONT_RIGHT

CAM_BACK_RIGHT CAM_BACK CAM_BACK_LEFT

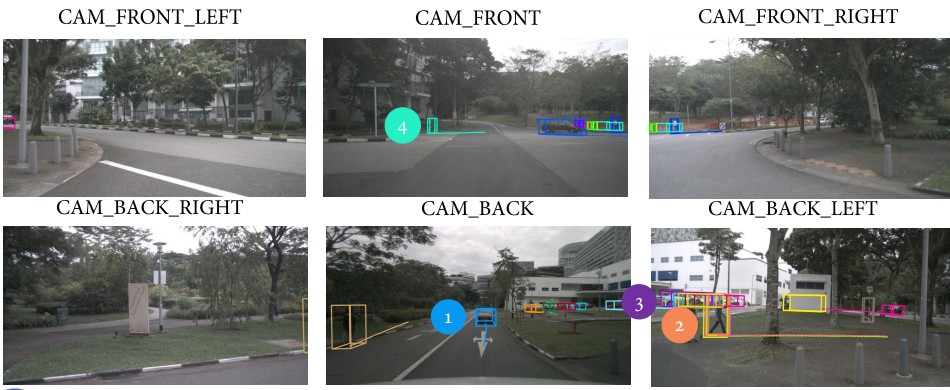

## 3D dense captioning

**Object 1**

**Pred**
black, shiny, and sleek car about 13 meters away in the back of ego car is moving quickly in the driving lane.

**GPT-4o**
Gray car about 20 meters away in the back of ego car is moving slowly in the driving lane.

**Gemini 1.5 Pro**
car about 15 meters away in the back of ego car is moving slowly in the driving lane.

**GT**
black, shiny, and sleek car about 13 meters away in the back of ego car is moving quickly in the intersection.

**Object 3**

**Pred**
black, shiny, and sleek car about 28 meters away in the back left of ego car is not moving.

**GPT-4o**
A person wearing dark clothing about 10 meters away in the back left of the ego car is walking on the sidewalk.

**Gemini 1.5 Pro**
car about 12 meters away in the back of ego car is moving slowly in the driving lane.

**GT**
white, silver, and black sleek about 27 meters away in the back left ego car is moving.

**Object 2**

**Pred**
person wearing a black shirt and blue jeans about 16 meters away in the back left of ego car is moving slowly in the walkway.

**GPT-4o**
A person wearing dark clothing about 10 meters away in the back left of the ego car is walking on the sidewalk.

**Gemini 1.5 Pro**
Person about 10 meters away in the back left of ego car is moving slowly in the walkway.

**GT**
person wearing a purple shirt and black jeans about 18 meters away in the back left of ego car is moving slowly in the walkway.

**Object 4**

**Pred**
person wearing a black shirt and black jeans about 36 meters away in the front of ego car is moving slowly.

**GPT-4o**
A person wearing dark clothing about 20 meters away in the front of the ego car is stationary on the sidewalk.

**Gemini 1.5 Pro**
car about 20 meters away in the front left of ego car is not moving in the driving lane.

**GT**
person wearing a white shirt and black pants about 36 meters away in the front of ego car is moving slowly.

Figure 6: **Qualitative results on TOD³Cap with GPT-4o and Gemini-1.5 outputs.**

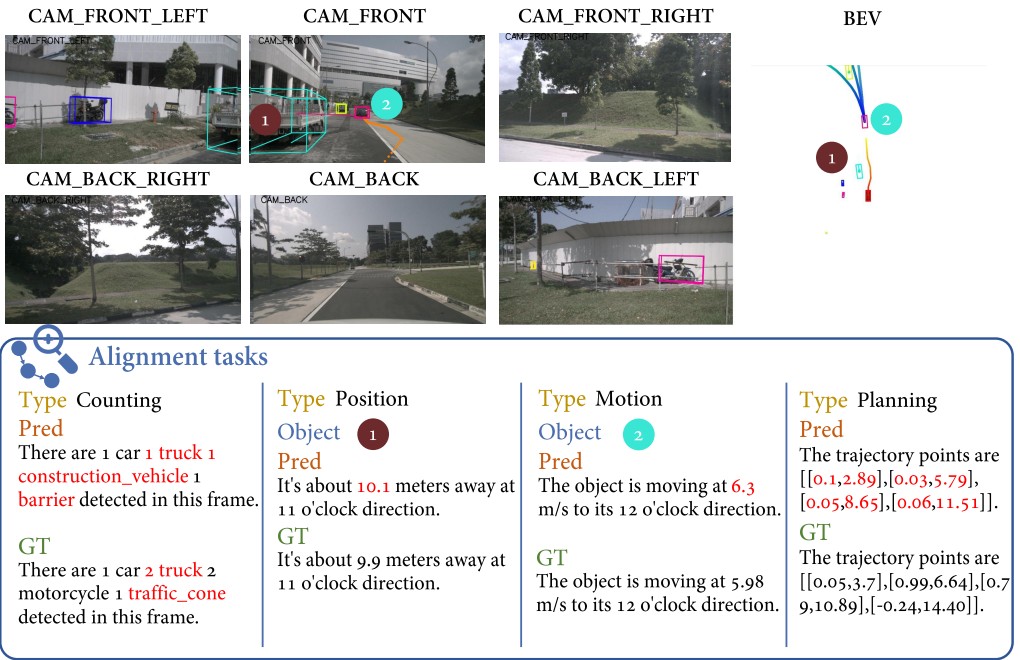

Figure 7: Qualitative results on alignment dataset.

## C.6 Caption evaluation

**GPT-3.5 for Logicality and Accuracy Assessment** The evaluation of visual language models for automated driving presented in this paper focuses on two main components: the logicality assessment of driving descriptions and the accuracy evaluation of driving behavior predictions.

For the logicality assessment of driving descriptions, we utilized GPT-3.5 to determine the correlation between driving behavior descriptions and the triggers for those behaviors in the model's predictions. Initially, we prompted GPT-3.5 to extract key information from each predicted description, including the car's driving behavior (movement and geographic location) and the causative factors (subject and form). We evaluated the associative relationship between the preceding and following events based on three criteria:

- The consistency between the car action implied by the causative factors and the predicted driving behavior.

- The logical coherence between the causative factors and the predicted driving behavior.

- The consistency between the details in the causative factors and those in the driving behavior.

Each metric was scored and summarized for each component.

Additionally, we conducted a comparative analysis of the similarity between predicted and manually labeled driving behaviors. Using GPT-3.5, we extracted key information, such as the car's action, purpose, and reason, from both the predicted and manually labeled driving behaviors for comparison. Each pair was scored based on the degree of match (complete, partial, or missing). The primary metrics for this comparison included:

- The similarity between driving behavior and geographic location.

- The objects and reasons leading to such driving behavior.

- Other relevant details.

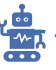

You are an expert system in autonomous driving. Help deal with these perception, prediction and planning tasks.

**Input prompt**

These are six photos taken from cameras positioned at the front, front-right, front-left, back, back-left, and back-right of the vehicle.
With picture urls of six photos of different views

or

These are five photos taken from cameras positioned at the front of the car in time series.
With picture urls of five front view photos, depending on the task.

Followings are several questions. Read the images above and answer the questions one by one.

**For Nu-X**

Describe the behavior of the ego car and the reason behind it.
Example 1 : Narration: the ego car smoothly transitions into the right lane on a bustling city street.
Reasoning: for the purpose of aligning itself with the traffic flow and entering the right lane.
Example 2 : Narration: the car is merging into the left lane.
Reasoning: because the lane is moving faster.
Example 3 : Narration: The ego vehicle gradually reduce its speed.
Reasoning: to safely navigate around the parked truck.
Example 4 : Narration: the car cautiously decelerates to a complete stop.
Reasoning: due to the fact that the intersection lying ahead is governed by a halt-inducing red light.
Example 5 : Narration: the car gradually decelerates, coming to a complete halt.
Reasoning: due to the presence of a red traffic light at the upcoming intersection.
Example 6 : the car meticulously maneuvers to the right lane.
Reasoning: so as to merge into the right lane due to upcoming

**For TOD³**

Describe the given object in the format of '<attribute> about <distance> meters away <localization> is <motion> <map>'.
Example 1 :  black, shiny, and sleek car about 25 meters away in the back of ego car is moving quickly in the driving lane.
Example 2 : person wearing a white shirt and black jeans about 26 meters away in the back of ego car is standing.
Example 3 : yellow bicycle about 46 meters away in the front of ego car is not moving.
Example 4 : white, silver, and gray car about 22 meters away in the front left of ego car is not moving.
Example 5 : person wearing a black shirt and black pants about 56 meters away in the front of ego car is moving slowly in the walkway.
Example 6 : trafficcone about 28 meters away in the back of ego car is not moving in the driving lane.

**For NuScenes-QA**

Answer each questions only in a single word or number, with out any extra output. Seperate answers with a comma.
Your output should be like '0,0,yes,no,no,no,car,car', where each answer corresponds to a question.
Example 1 : moving
Example 2 : without rider.
Example 3 : pedestrian.
Example 4 : yes.
Example 5 : 2.
Example 6 : bicycle.

**For Command**

Please predict the control signal of the ego car. Choose from: 'Turn right', 'Turn left', and 'Forward'.
Then predict the motion. Only give the final control signal in one sentence.
Example 1 : Forward and keep speed.
Example 2 : Forward and stationary.
Example 3 : Forward and increase speed.
Example 4 : Forward and decrease speed.
Example 5 : Turn right and keep speed.
Example 6 : Turn left and keep speed.

Figure 8: **Prompts for GPT-4o and Gemini-1.5.**

Penalties were applied for instances of fictitious information, which reduced the total score. The mean of these scores constituted the scene description score. The total prompt for GPT-3.5 is shown in Fig. 9

**System prompt**

    You are a smart and strict logical evaluator with extensive driving experience, and try your best to give reasonable and logical evaluation result.

**Input prompt**

    Analysis adaptations of output, the main scoring principles are:
1. Consistency between the driving behavior and reasoning of output (0, max:5).
2. Coherence and logic between the driving behavior and reasoning of output (0, max:5).
3. Consistency of detail between the behavior and reasoning of output (0, max:5).
4. Driving behavior and location (0, 5, 10, 15, max:20).
5. What and why cause the driving behavior (0, 5, 10, max:15).
6. Other details (0, 5, max:10).
7. Presence of irrelevant information (-5, max:0).

    You need provide the rigorously evaluated score for each of the criteria. Ensure the fairness of the output results.

**Response**

Figure 9: Prompt for GPT-3.5 on semantic evaluation

# D    More on Nu-X

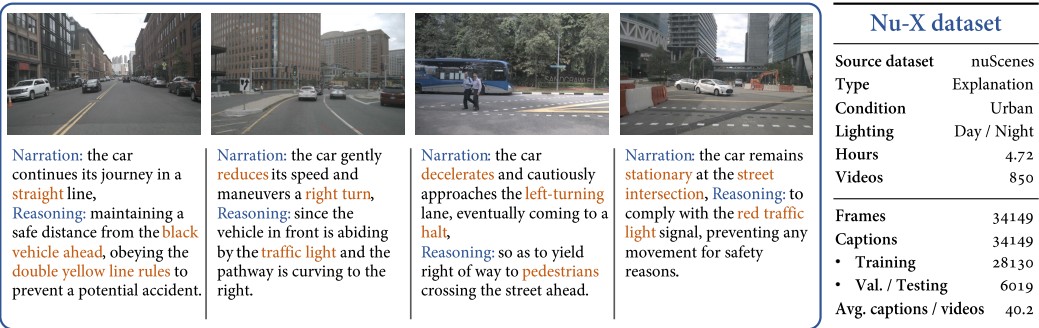

| Nu-X dataset | |
| --- | --- |
| Source dataset | nuScenes |
| Type | Explanation |
| Condition | Urban |
| Lighting | Day / Night |
| Hours | 4.72 |
| Videos | 850 |
| Frames | 34149 |
| Captions | 34149 |
| • Training | 28130 |
| • Val. / Testing | 6019 |
| Avg. captions / videos | 40.2 |

Figure 10: Examples of Nu-X dataset.

## D.1    Annotation procedure

To balance accuracy and effectiveness of annotation, we conducted a process incorporating both human labelling and LLM diversification (see Fig. 11). The training / validation branches of nuScenes contain 850 videos, each spanning about 20 seconds, with approximately 40 key frames per video annotated. labelling all frames is inefficient as driving behaviors and road conditions in adjacent frames are often similar. Each video typically features only 3-4 distinct driving behaviors. However, identical annotations across frames risk over-fitting, reducing effectiveness. Therefore, we used MLLM to diversify expressions and add scene-description details to each frame. The annotation process includes the following procedures:

**Human labelling**    We employed five professional annotators for a total of 126 hours. All annotators are familiar with US driving rules and have driving experience. They were instructed to describe specific driving behaviors and the potential reasons for these decisions. Annotations for each video include the time intervals of certain driving behaviors, along with narration and reasoning.

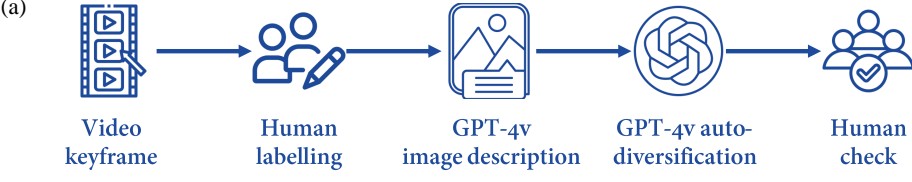

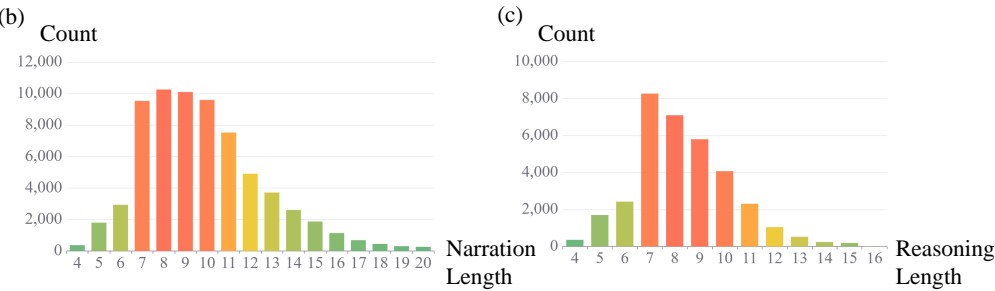

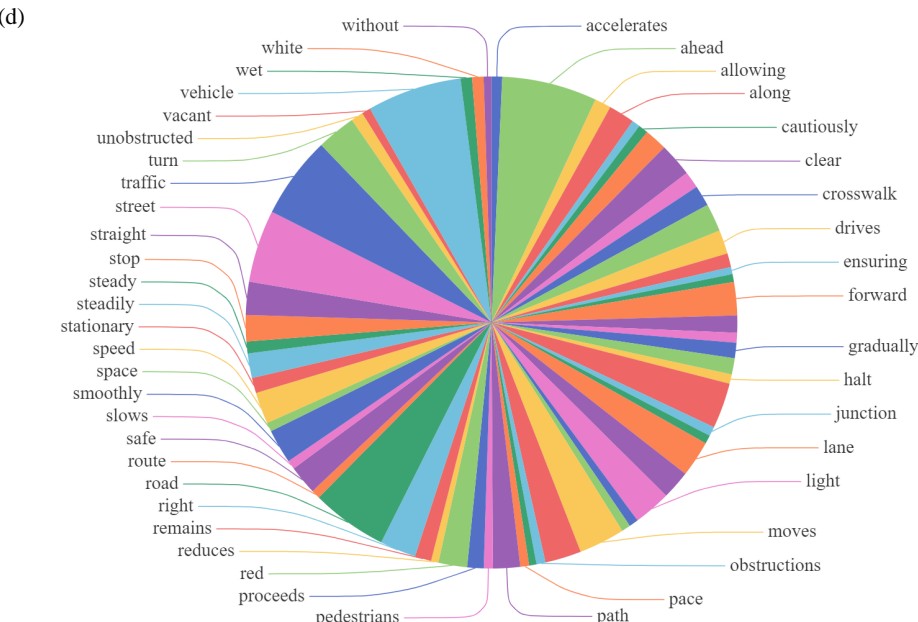

Figure 11: Annotation and data statistics of Nu-X. (a) The annotation process of Nu-X dataset; (b) Distribution of the sequence length of narration; (c) Distribution of reasoning sentence length; (d) Distribution of different vocabulary in Nu-X.

For narration, annotators focused on at least 11 driving behaviors, such as directional decisions (e.g., go forward, turn left, turn right, U-turn) and velocity commands (e.g., accelerate, decelerate, stop, stay stationary). Descriptors like "slightly" or "sharply" were encouraged.

For reasoning, annotators incorporated specific road conditions, key instances (e.g., pedestrians, front cars), and traffic rules (e.g., traffic lights, stop signs, yellow lines).

**GPT-4v image description**   Before diversification process, we firstly leverage GPT-4v to generate the image description of multi-view camera images to extract the visual information. Specifically, we provide GPT-4v with each frame of multi-view images in nuScenes and task it with capturing and recording key traffic information from these images. This information includes, but is not limited to, pedestrians, vehicles, traffic signals, and other relevant road signs. Through this process, GPT-4v is able to automatically extract and provide detailed descriptions of the elements and dynamics present in each image frame in a sentence of about 120 word length.

**GPT-4v auto-diversification**   With the image description, we utilize GPT-4v to automatically generate diverse outputs based on the human-labeled captions, which is particularly useful for creating varied descriptions and perspectives of the same set of images. We also constraints the text output to follow the original main driving behavior and reasoning of human annotators to ensure the correctness.

**Human check**   We conduct a thorough human check to ensure accuracy and relevance after the auto-diversification. This step involves experts reviewing and validating the multiple interpretations generated by GPT-4v, verifying that the descriptions accurately reflect the content of the images. The human check process is essential for maintaining high standards of quality and reliability in our traffic analysis, ensuring that the diversified outputs align with real-world observations.

### D.2   More on data statistics

We show some key results of data statistics in Fig. 11(b)-(d). The average length of a narration is 8.29 words for a sentence, and the average length of reasoning is 11.25 words.

## E   More on command dataset

The ground truth of the nuScenes dataset provides only three types of directional commands: <TURN LEFT>, <TURN RIGHT>, and <FORWARD>, which lack the necessary velocity information. To address this, we labeled a command dataset using programmed protocols based on the ground truth future steps, incorporating additional velocity commands such as <ACCELERATE>, <DECELERATE>, <STATIONARY>, and <KEEP SPEED>.

Directional commands were generated by analyzing the trajectory slope, applying a criterion where driving 5 meters with only 1 meter of lateral drift was considered.

For velocity commands, polynomial fitting was used to smooth the speed curve derived from the average speed between predicted points. The average acceleration over the next 4 seconds was then used to determine the velocity commands. Given the vehicle's increased sensitivity to speed changes when approaching zero, different thresholds for speed changes were set: $0.36\,\text{m/s}^2$, $0.12\,\text{m/s}^2$, and $0.05\,\text{m/s}^2$ for initial speeds of >2 m/s, 1-2 m/s, and <1 m/s, respectively.

