# OpenReview forum: "Hint-AD: Holistically Aligned Interpretability in End-to-End Autonomous Driving"
_robot-learning.org/CoRL/2024/Conference — CoRL 2024_

### Official Review · Reviewer_3Cfu · 2024-07-11
**Aligning an autonomous driving system with multimodal LLM to improve interpretability**

**Originality:** 3
**Technical Quality:** 2
**Clarity Of Presentation:** 2
**Potential Impact:** 2
**Recommendation:** 3
**Confidence:** 4

**Review:**

Strengths:
The paper shows that aligning the MLLM and AD models can provide better results for driving explanation, 3D dense captioning, and command prediction. The proposed method has been proven effective on two popular end-to-end AD models, UniAD and VAD.

The paper also provided dedicated data called Nu-X, extended from the nuScenes dataset for driving explanation.

Weaknesses:
The presentation of the method lacks a certain level of detail. The pipeline figure (Fig. 2) is very compact and does not provide much detailed information about the core component, the mixer sub-network. There should be a detailed figure to make it clearer.

What are the loss functions to optimize the model in different training stages?

In section 3.5, it is not clear why the MLLM parameters are frozen or whether they are fine-tuned.

The ablation studies in Tables 2 and 3 are not comprehensive. First, each time, only one module is removed. What about the performance when removing more than one of the proposed modules together? Second, are the ablation studies consistent on Hint-VAD?

The paper emphasizes interpretability, but interpretability is never clearly defined in this paper, and how to measure the quality of interpretability is also not clear. A related question concerns the Nu-X dataset for driving explanation. The authors never mentioned how good the annotation scheme of Nu-X is regarding "interpretability."

Some minor issues/errors:

In Fig. 2, layers are subscripted by "early" and "late," but in the text, they are subscripted by "front" and "end."

"MLLM" is never defined before it appears.

There is an error in Table 2; in the TOD2Cap-M column, 48.5 should be boldfaced instead of 48.0.

One more thing, which may be off the focus of this paper: I wonder whether the alignment also has some impact on the performance of the AD tasks.

**Quality Of The Limitations Section:**

2

**Questions For Rebuttal:**

A more comprehensive ablation study is needed to validate the model's performance.

A better definition of interpretability is helpful to understand the contribution of the proposed approach.

**Robotics Focus:**

3

**Summary Of Paper:**

This paper proposes a mixer sub-network to align the intermediate outputs from a BEV perception-prediction-planning system for autonomous driving (AD) with a multimodal LLM (MLLM) for driving explanation, 3D dense captioning, and command prediction.  The contribution of this work is to enhance interpretability in end-to-end AD, achieving better performance than the declarative interpretability that is not grounded in the intermediate outputs of AD systems.

**Summary Of Recommendation:**

The paper shows some potentials for interpretable autonomous driving using multimodal LLM, but the lack of detailed insights of the model's design and evaluation result in a week rejection.

---

### Official Review · Reviewer_dfFt · 2024-07-22
**Multi-modal language model  for dynamic driving scenes and first annotation data on NuScenes**

**Originality:** 3
**Technical Quality:** 4
**Clarity Of Presentation:** 4
**Potential Impact:** 3
**Recommendation:** 3
**Confidence:** 4

**Review:**

This paper introduces Hint-AD, a method that provides aligned interpretability in autonomous driving scenarios leveraging multi-modality language models(MLLMs). The method can be thought of a variant of 3D-LLM but with autonomous-driving-specific modalities/tokens including features for BEV, tokens for past and future predicted trajectories and tokens for motion planning.

To incorporate those changes, the authors introduce certain architectures named "holistic token mixer" designs to fuse the information together with multi-head-self-attention and multi-head-cross-attention. To input those driving-related context information into the MLLM finetuning, the authors introduced "Barbell adaptation" that only changes the parameters for the first two and last two layers of the network to avoid over-reliance on language tokens and ignores context information.

Another contribution is the introduction of Nu-X dataset, which is the first language annotation effort on NuScenes, a prominent dataset in autonomous driving. Following common recipes for MLLM training, Nu-X also introduces 4 different tasks including counting, position, motion and planning.

Overall, I believe the idea of "Aligned interpretability" is a better alternative of "Declarative interpretability" by incorporating driving-specific immediate information into the explanation.  I feel that this architecture can be even extended beyond just interpretability research, but in the future, directly produces better track prediction and planning signals. Although interestingly, the current results only support better scene understanding, not particularly strong in terms of numeric tasks like counting or planning (see figure 3). It could be a future direction.

**Quality Of The Limitations Section:**

3

**Questions For Rebuttal:**

The experiment section could be made stronger:

1. Baselines in table 1 could be extended. For example, although many closed-source VLMs do not support finetuning, they could still be used as baseline including GPT-4V, Gemini-1.5. Other 3D-LLMs that are good at spatial metric prediction including SpatialRGPT, Spatial GPT, Transcrib3D can also be considered adding as baselines.

2. While table 4 investigate the effectiveness of "Barbell adaption" against other approaches in terms of layer-choosing strategies, it would also be helpful evaluate against other adaption strategies beyond layer-choosing, for example, LoRA, DoRA, IA3.

**Update after rebuttal:**
Thank authors for the additional experiments. I believe they made the paper stronger. I have raised my rating on technical quality.

**Robotics Focus:**

4

**Summary Of Paper:**

Multi-modal language model for aligned interpretability in autonomous driving

**Summary Of Recommendation:**

Good extension of 3D-LLM s to dynamic scenes in autonomous driving as well as the contribution of a useful dataset

---

### Official Review · Reviewer_rKXi · 2024-07-23
**The authors present a well motivated approach that is extensively evaluated across various benchmarks and metrics and against several methods. The proposed work provides strong technical contributions and the technical decisions made throughout the paper have been explained with clarity.**

**Originality:** 3
**Technical Quality:** 4
**Clarity Of Presentation:** 3
**Potential Impact:** 3
**Recommendation:** 4
**Confidence:** 3

**Review:**

The work presented by the authors is well motivated and clearly explained throughout the paper. The authors have made strong technical contributions in both designing a system to improve interpretability and also releasing a dataset towards the same.

Strengths:
- The authors present the motivation for the work clearly.
- Claims made by the authors are supported through extensive quantitative evaluations, while also evaluating the results qualitatively. Evaluations are made on a wide variety of metrics, datasets and methods.
- Architectural decisions made in the paper (and the reasons behind them) are conveyed clearly and are well justified through ablations.
- The proposed approach is evaluated against several SOTA approaches and shows strong potential.

Weaknesses:
- It is important to consider inference time and model size when building systems for autonomous driving as this ultimately dictates the applicability of a method. Though the authors provide inference time for different batch sizes for the AD system and language decoder, it would be beneficial to also include latency measurements for the entire system.
- The limitations of the proposed approach have not been covered by the authors.
- The authors present a claim that demonstrating their proposed approach on UniAD and VAD demonstrates the methods' generality. However, it is unclear this is the case. It would be beneficial to provide some clarity on the generalizability of the proposed approach.
- Though it may be implied, it would be helpful to mention that F'_{plan} is projected to D_{embed} for clarity in the methodology section. This is a minor point.

**Quality Of The Limitations Section:**

3

**Questions For Rebuttal:**

1. What is the latency of the proposed approach as a whole i.e., of the augmented AD system?
2. What are the limitations of the proposed approach?
3. How does the success of the proposed approach on UniAD and VAD demonstrate the methods' generality? (Claim made in the paper)

**Robotics Focus:**

3

**Summary Of Paper:**

The paper presents a systems level approach to improving interpretability in Autonomous Driving (AD) systems. The authors argue for an aligned interpretability approach as opposed to declarative interpretability in order to improve interpretability of the decision making process within AD systems. Specifically, they propose the use of intermediate outputs (from perception, prediction and planning) in the AD stack to provide contextual information to a foundation model to achieve this goal. They propose the use of features extracted from multi-view images as well as track, motion, and planning information obtained from the rest of the AD stack and present a method to efficiently and effectively combine this contextual information towards generating better aligned responses. Lastly, they created a dataset by augmenting the nuScenes dataset with detailed contextual information.

**Summary Of Recommendation:**

The authors have presented their approach and its motivation with clarity and have extensively evaluated it both quantitatively against other methods, across several metrics and benchmarks as well as qualitatively. The paper, however, would benefit from clarifying claims made around generality of the approach and from providing additional information around the limitations of the system. That being said, the proposed approach provides sufficient contributions for me to recommend a Weak Accept.

---

### Official Review · Reviewer_kWqV · 2024-07-23

**Originality:** 4
**Technical Quality:** 4
**Clarity Of Presentation:** 4
**Potential Impact:** 3
**Recommendation:** 3
**Confidence:** 4

**Review:**

This paper does an excellent job of motivating why aligned interpretability is necessary (i.e., the alternative is post-hoc "justification" at best), demonstrating through disalignment metrics the drawbacks of purely declarative approaches to interpretability, and supporting their design choices through impressively thorough quantitative ablation studies. I am pleased to recommend its acceptance, however there are some questions about the limitations of this work which, if discussed, would help readers better assess the impact of this work:

(1) Lingering disalignment: While processing intermediate outputs through the holistic token mixture model enables the MLLM to access more information, and therefore to provide better grounded explanations, alignment is still treated as an emergent property of training rather than enforced by design. This is evident in the disalignment metrics where, although significantly improved, Hint-AD still displays some degree of mismatch between module outputs and language explanation. How may this be remedied? Is greater model/data scale all that is needed to drive disalignment down to 0? An illustrative comparison might be to contrast this approach to tool-using LLMs, i.e., LLMs that in order to improve QA performance can query specialist models (e.g., planning, prediction, perception modules) and directly include their output in the response.

(2) Discussion of non-modular approaches to end-to-end AD: In this work "end-to-end" refers to "end-to-end differentiable" AD systems, while to some researchers and practitioners it has an alternative connotation of end-to-end "black box" neural network policies that do not have clear submodules. The authors do make it clear that Hint-AD can only be applied to "any end-to-end AD system that decomposes AD into perception, prediction and planning", but many black box policies do train additional output heads (with the claim that this is done "for interpretability") that provide similar perception, prediction, and planning outputs but notably are not "intermediate" in the sense that they are not on the computational path for decision making. One might imagine developing a system similar to Hint-AD for an end-to-end black box AD system with the output of these auxiliary heads. Would such an AD-language system still fall in the category of declarative interpretability? Do the authors believe that aligned interpretability can ever be achieved with models with a black box backbone?

**Quality Of The Limitations Section:**

2

**Questions For Rebuttal:**

From the comments/questions above, please prioritize:
- Is greater model/data scale all that is needed to drive disalignment down to 0?
- Could aligned interpretability be achieved for black-box models?

**Robotics Focus:**

3

**Summary Of Paper:**

In this paper the authors define a distinction between "declarative" and "aligned" interpretability, arguing that aligned interpretability, where language explanations are grounded in the internal state of a system, is necessary to explain _why_ certain decisions are made. This motivates the authors' development of the Hint-AD framework which provides aligned interpretability for modular perception-prediction-planning autonomous driving stacks (specifically UniAD and VAD in this work), and moreover results in improved performance on various vision-language driving tasks including a new human-labeled driving explanation dataset + task Nu-X. These advances are enabled by a new AD-language model architecture that incorporates intermediate module outputs and is trained on various online alignment tasks to encourage the learning of token representations well-adapted to interpretable reasoning.

**Summary Of Recommendation:**

I recommend this paper's acceptance at CoRL 2024 as it both presents strong benchmark results/corresponding insights for AD practitioners, in addition to addressing a topic of broader interest/importance in the robot learning community today: the importance of alignment in interpretable AI. However, as noted above, I believe the paper would be improved if the limitations of this work could be more clearly established.

---

### Official Review · Reviewer_Kmwf · 2024-07-24
**Hint-AD: Holistically Aligned Interpretability in End-to-End Autonomous Driving**

**Originality:** 3
**Technical Quality:** 2
**Clarity Of Presentation:** 4
**Potential Impact:** 3
**Recommendation:** 3
**Confidence:** 3

**Review:**

## Quality and Clarity


Strengths:

-  The methodology is thoroughly explained, with clear diagrams illustrating the HINT-AD architecture.
- Clear explanation of the annotation pipeline and benchmarking, including the interoperability and alignment.

Weaknesses:

- Some technical details, such as the specifics of the rationale for selecting adaptors over SFT, the cross-attention mechanism between the learnable and instance tokens, and masking techniques, are not fully explained, which could hinder reproducibility.
- The paper could benefit from a more detailed discussion of the limitations of the approach.

## Originality


Strengths:

- The token mixer module and the alignment task are innovative in robotics applications.
- The Nu-X dataset addresses a gap in existing autonomous driving datasets.

Weaknesses:

- It builds upon existing work in multi-modal language models such as LLAVA and autonomous driving systems. A more explicit discussion of how this work advances beyond previous approaches would strengthen the paper.

## Significance


Strengths:

-  The HINT-AD framework demonstrates improvements over baseline models.
-  The approach could enhance the explainability of autonomous driving systems.
-  The Nu-X dataset introduced could facilitate further research in this area.

Weaknesses:

- The real-world applicability, impact of the latency from the token generation and scalability of the approach are not thoroughly discussed.
- Authors have not shown how MLLMs such as GPT4 or gemini1.5 perform compared to the proposed methods.

**Quality Of The Limitations Section:**

1

**Questions For Rebuttal:**

- What is the rationale for not passing planning and ego motion through an attention layer similar to the BEV and instance blocks?
- Explain the why not perform a SFT instead of adaptor, give the task performed is different compared to typical training data in the LLM ?
- Given that MLLM adapter training (e.g., LoRA) is shown to preserve pretrained information and language understanding, why weren't low-rank adapters used instead of full adapters?
- It is highly recommended to include comparisons with multi-modal language models such as GPT-4, LLAVA-v1.6, and Gemini 1.5 for VQA and other tasks. This comparison would help quantify the improvement over out-of-the-box models, especially given that LLAVA is the base MLLM for HINT-AD.
- The declarative alignment results of the HINT-AD models proposed here should be compared with other declarative alignment models. This could highlight whether HINT-AD is increasing overall efficacy of the alignment or specifically improving the efficiency of aligned interpretability.

**Robotics Focus:**

3

**Summary Of Paper:**

In this paper, the authors introduce a novel approach called "aligned interpretability" for autonomous vehicle (AV) actions. This algorithm/method aims to provide intermediate interpretability to the AV decision-making processes by aligning language generation with the intermediate outputs of the AD model.  Authors have developed a new interpretability method, introduced a new dataset (Nu-X), and created a framework (HINT-AD) that generates language aligned with the perception-prediction-planning outputs of the AD model.

**Summary Of Recommendation:**

I recommend a weak accept for this paper. The work presents a novel and potential impactful contribution to the field of interpretable AI with the introduction of the HINT-AD framework and the concept of aligned interpretability. The paper demonstrates clear methodological rigor and fair experimental validation, showing some improvements over baseline models across multiple language tasks. The creation of the Nu-X dataset is also a valuable contribution to the research community and will help advance the field. However, the paper has some limitations that prevent a stronger recommendation. These include the 1. Need for more detailed technical explanations in some areas, 2. A lack of comparison with state-of-the-art multi-modal language models, and 3. Insufficient discussion of real-world applicability such as latency. Despite these shortcomings, the  approach and potential impact of this work make it worthy of inclusion in the conference, with the expectation that the authors will address these issues in the final version or in future work.

---

### Official Review · Reviewer_HNsL · 2024-07-25
**Interesting work, decent contributions, but future impact is uncertain**

**Originality:** 3
**Technical Quality:** 4
**Clarity Of Presentation:** 3
**Potential Impact:** 3
**Recommendation:** 3
**Confidence:** 3

**Review:**

Praise:
- The paper addresses a critical aspect of interpretability in ADAS systems' behaviors.
- An extensive ablation study across three datasets evaluates the impact of various alignment strategies, the design of a holistic token mixer, and the placement of adaptation injections within the architecture.
- The authors contribute the Nu-x driving explanation dataset, which promises to foster further advancements in model interpretability research.

Concerns and Questions:
- A primary concern is the fundamental approach of the system: as holistic pipelines gradually give way to end-to-end planners, the reliance on perception and prediction outputs becomes problematic. Hence, these trends might strongly limit the potential impact of this work on the future of AV development.
- Another drawback is the system's dependence on specific token formats, potentially limiting its applicability. In holistic pipelines, outputs from perception and prediction systems are tailored to the planner’s requirements, undergoing continuous evolution throughout AV stack development. This necessitates frequent re-training of the proposed DNN with each pipeline modification.

**Quality Of The Limitations Section:**

2

**Questions For Rebuttal:**

Please, see the review section.

**Robotics Focus:**

3

**Summary Of Paper:**

The paper introduces a holistically aligned AD-language system designed to generate explanations for driving behaviors. Specifically, the authors demonstrate that by integrating tokens and employing an alignment procedure across all components of the holistic perception-prediction-planning stack, the LLM model enhances performance in driving language tasks. The effectiveness of this approach is validated on three distinct datasets, including the newly introduced Nu-x dataset, which provides driving explanations aligned with frames from the widely used nuScenes dataset.

**Summary Of Recommendation:**

Interesting work, decent experimental session with good ablation studies. But the core of the approach brings questions about future impact of this work. Hence I propose "weak accept"

---

### Author Rebuttal · Authors · 2024-08-13

We would like to extend our heartfelt thanks to all the reviewers for their insightful and constructive feedback. Your recognition of our contributions on exploring MLLM for the aligned interpretability in autonomous driving (AD), the contribution of Nu-X dataset, and extensive evaluations is greatly appreciated.

Here we provide a summarization on common issues, **an attachment file of supplements and revised paper is presented**.

> **Future Value and Trends:** Concerns are expressed about the future value of modular approach, particularly in the trend towards fully end-to-end methods in AD.

Hint-AD is built on end-to-end AD systems (UniAD and VAD) that incorporate intermediate outputs for improved performance and interpretability. Incorporating intermediate outputs remains a beneficial and common practice in state-of-the-art AD systems. Recent high-performance AD systems often utilize intermediate outputs to enhance reliability and transparency (detailed in reply to reviewer HNsL).

> **Comparison with GPT-4o and Gemini:** Reviewers suggest including comparisons with state-of-the-art multimodal language models such as GPT-4 and Gemini.

We compared Hint-AD with GPT-4o and Gemini 1.5. The results showed that Hint-AD significantly outperforms these models, particularly in grounding tasks such as 3D dense prediction and command prediction. For quantitative and qualitative details, please refer to Section 3 of the attachment.

> **Selection of Adapter over Other Tuning Approaches:** Reviewers questioned the choice of adapters over other parameter-efficient fine-tuning methods like LoRA.

Adapters were chosen for their stability and efficiency in training. Comparative analysis shown in Section 2.2 in the attachment shows that adapters outperform other methods like LoRA and DoRA, ensuring stable training and superior overall performance.

> **Limitations of Our Approach:** A more detailed examination of the limitations of the approach is recommended.

Limitations of Hint-AD include:
- Due to its pipeline-specific nature, changes in the intermediate output format require modifications in the design of the token mixer. For purely end-to-end models, i.e. the black-box models, adjustments must be made to handle latent outputs.
- The LLaMA-based language decoder is relatively time consuming, taking about 1 second to inference. More investigations on smaller size models alternatives, i.e. MiniChat-1.5-3B, StableLM-3B-4E1T.

---

### Decision · Program_Chairs · 2024-09-04

**Decision:**

Accept

**Comment:**

Strengths:
1. Contributes a dataset (nu-X) for explainability in autonomous driving
2. Extensive quantitative and qualitative evaluations
3. Modular approach to interpretability, a key safety research area

Weaknesses:
1. Modularity for the sake of interpretability may not survive the larger trend of end to end methods taking over. Future value of this work is contentious.
2. Please cover limitations in more detail:
- how well does the method generalize
- lack of enforced alignment
- comparison against multimodal LLMs
- latency of system for driving


Reviewer HNsL:The reviewer questioned the future value of the modular approach in Hint-AD, given the trend towards fully end-to-end methods in autonomous driving. They also pointed out the system's dependence on specific token formats, which could limit its applicability. The authors clarified that Hint-AD is already built on end-to-end AD systems and that incorporating intermediate outputs is a common practice in such systems. They also highlighted that Hint-AD is not dependent on specific token formats and can work with various types, given sufficient data and a well-designed encoder.

Reviewer Kmwf: The reviewer raised several concerns, including a lack of technical details in certain areas, the absence of comparison with state-of-the-art multi-modal language models, and insufficient discussion of real-world applicability, such as latency. The authors provided more detailed technical explanations, included comparisons with GPT-4 and Gemini 1.5, and added a discussion on real-world applicability, focusing on latency and inference speed.

Reviewer kWqV: The reviewer questioned the limitations of Hint-AD, particularly regarding lingering disalignment and the applicability to non-modular, black-box end-to-end AD systems. The authors discussed the possibility of reducing disalignment through strategic model design and larger data scales. They also addressed the applicability to black-box models, explaining that aligned interpretability could be achieved by passing BEV perception and inner states to the MLLM.

Reviewer dfFt: The reviewer suggested extending the baselines in the experiments and evaluating other adaptation strategies beyond layer-choosing. The authors included additional experimental results featuring GPT-4, Gemini 1.5, LoRA, and DoRA. They also explained why most 3D-LLM baselines are incompatible with their use case.

Reviewer 3Cfu: The reviewer pointed out a lack of clarity in the method presentation, the ablation studies, and the definition of interpretability. The authors provided a more detailed figure, clarified the loss functions and training stages, expanded the ablation studies, and provided a clearer definition of interpretability and how it's measured.